# Feasibility of a Novel Therapist-Assisted Feedback System for Gait Training in Parkinson’s Disease

**DOI:** 10.3390/s23010128

**Published:** 2022-12-23

**Authors:** Carla Silva-Batista, Graham Harker, Rodrigo Vitorio, Fay B. Horak, Patricia Carlson-Kuhta, Sean Pearson, Jess VanDerwalker, Mahmoud El-Gohary, Martina Mancini

**Affiliations:** 1Department of Neurology, Oregon Health & Science University, Portland, OR 97239-3098, USA; 2Department of Sports, Exercise and Rehabilitation, Northumbria University, Newcastle upon Tyne NE1 8ST, UK; 3APDM Wearable Technologies—An Clario Company, Portland, OR 97239-3098, USA

**Keywords:** Parkinson’s disease, visual feedback, wearable technology

## Abstract

We tested the feasibility of one session of treadmill training using a novel physical therapist assisted system (Mobility Rehab) using wearable sensors on the upper and lower limbs of 10 people with Parkinson’s disease (PD). Participants performed a 2-min walk overground before and after 15 min of treadmill training with Mobility Rehab, which included an electronic tablet (to visualize gait metrics) and five Opal sensors placed on both the wrists and feet and on the sternum area to measure gait and provide feedback on six gait metrics (foot-strike angle, trunk coronal range-of-motion (ROM), arm swing ROM, double-support duration, gait-cycle duration, and step asymmetry). The physical therapist used Mobility Rehab to select one or two gait metrics (from the six) to focus on during the treadmill training. Foot-strike angle (effect size (ES) = 0.56, 95% Confidence Interval (CI) = 0.14 to 0.97), trunk coronal RoM (ES = 1.39, 95% CI = 0.73 to 2.06), and arm swing RoM (ES = 1.64, 95% CI = 0.71 to 2.58) during overground walking showed significant and moderate-to-large ES following treadmill training with Mobility Rehab. Participants perceived moderate (60%) and excellent (30%) effects of Mobility Rehab on their gait. No adverse events were reported. One session of treadmill training with Mobility Rehab is feasible for people with mild-to-moderate PD.

## 1. Introduction

Gait impairments are common in people with Parkinson’s disease (PD) and often lead to falls [1]. People with PD have a gait pattern characterized by shuffling of the feet and low foot clearance due to reduced foot-strike angle, reduced ankle plantar flexion at toe-off, and reduced stride length when compared to healthy controls [2,3,4]. The movement of the trunk and arms is crucial to dynamic balance during walking [5,6]; however, people with PD have decreased range-of-motion (RoM) of the trunk and arms compared to healthy controls [3,7], that is associated with an increased risk of falls [5,6]. These deficits in both upper and lower body gait metrics, which are usually multifactorial in origin [8], require a comprehensive assessment to identify the risk of falling and to target intervention for people with PD.

Many approaches to treat gait impairments exist and can be effective [9] but need to be paired with a comprehensive screening approach [9]. The physical therapist (PT) observes patients’ walking patterns and provides verbal and/or somatosensory feedback to improve their patients’ mobility. Although this approach allows patients to become more aware of how they move so they can correct any abnormal gait strategies, this method is not optimal [9]. In fact, gait observation is subjective, depends on the expertise of the PT and might be inaccurate. An objective characterization of gait impairments that are difficult to observe, such as a reduced trunk motion or decreased foot clearance would allow the PT to provide patient-specific and precise gait feedback during screening to determine potential interventions.

Wearable sensor-based systems can be used by the PT to provide feedback, in real-time, based on objective gait metrics [10]. Feedback-based interventions, using wearable sensors have shown promising results for gait rehabilitation [10]. Current, commercially available, gait biofeedback systems are all exclusively based on lower-body metrics that do not reflect dynamic balance control of the upper body [10,11]. This is the case for the gait trainer (Biodex Medical Systems, Inc., Shirley, NY, USA) that must be on a treadmill and uses only step length, speed and symmetry for gait training. In addition, the GaitSense 2000 (LiteGait Mobility Research, Tempe, AZ, USA) also uses an instrumented treadmill to measure swing phase duration and step length. The portable SmartStep system (Andante Medical Devices, Inc., White Plains, NY, USA) can be used overground and measures the force of the heel during gait using a force-sensing insole inside the patient’s shoe. Lastly, the WalkJoy system (WalkJoy, Long Beach, CA, USA) is a tactile biofeedback system that contains a small computer and a gyroscope unit attached to the shank below the knee, to measure the angle and speed of leg movement. As the foot strikes the ground, WalkJoy provides a vibrating sensation to the nerves below the knee. The system is limited to measuring leg movement, and provides no spatial metrics and only limited temporal metrics of gait.

Mobility Rehab, a novel, PT-assisted visual feedback system, was developed as a prototype [12] for providing real-time measures of arm swing and trunk lateral stability in addition to spatial and temporal gait metrics of the lower body, that are very important for good gait in people with PD. The Mobility Rehab system uses wireless, wearable inertial sensors (Opals, APDM Wearable Technologies—A Clario company) on the wrists, feet, and lumbar to improve the accuracy and effectiveness of PT’s feedback to his/her patients by providing objective measures of upper and lower body gait metrics in real-time. Thus, Mobility Rehab is the first commercial system for visual biofeedback that allows the PT to select among a variety of upper and lower body gait metrics.

Our aim was to test the feasibility of one session of treadmill gait training with Mobility Rehab in people with PD and whether one session could modify gait metrics during overground walking. A meta-analysis has demonstrated small-to-moderate effect size (ES) on gait metrics (e.g., walking speed, stride and step length) after short term (e.g., one month) of gait-specific training (e.g., treadmill training) in people with PD [13]. Our previous rehabilitation study [14] showed that 6 weeks of Agility Boot Camp with Cognitive Challenges Intervention resulted in small to large effects sizes (ES) on foot-strike angle (ES = 0.45), trunk coronal RoM (ES = 0.45), and arm swing RoM (ES = 0.95) in people with mild-to-moderate PD. Here, we hypothesized that one session of treadmill gait training with Mobility Rehab would show small to large ES on overground upper and lower body gait metrics in people with mild-to-moderate PD. We also investigated whether changes in upper and lower body gait metrics were associated with each other.

## 2. Materials and Methods

### 2.1. Participants

Ten individuals with PD were diagnosed by movement disorders specialists as having idiopathic PD based on the United Kingdom Brain Bank criteria [15]. All people with PD were eligible if they: (a) were older than 60 years of age; (b) had mild to moderate PD (Hoehn and Yahr Levels II–III); (c) were on stable antiparkinsonian medication; (d) were able to stand and walk for at least 2 min without an assistive device to make sure that they could perform the walking trials; and (e) had been referred to Oregon Health & Science University (OHSU) physical therapy for gait rehabilitation. Exclusion criteria were: dementia, stroke or any medical conditions that could prevent participation (e.g., major musculoskeletal, sensory loss, visual and auditory problems). People with PD were assessed in the practical OFF levodopa state (12 h withdrawal of antiparkinsonian medication). All subjects gave written informed consent in accordance with the Declaration of Helsinki. The protocol was approved by the Oregon Health & Science University (OHSU) Institutional Review Board (IRB) (#16282).

### 2.2. Experimental Design

This was a feasibility study (convenience sample) of one visit to verify the immediate effects of a novel, PT-assisted visual feedback system. Ten individuals with PD performed a 2-min walk overground (comfortable pace along a 9-m corridor) before and after 15 min of treadmill gait training with Mobility Rehab in the Balance Disorders Laboratory at OHSU. All individuals with PD were tested in the “OFF” state, after at least a 12-h overnight withdrawal from anti-parkinsonian medications.

#### 2.2.1. Overground Walking

Overground walking was performed before and after 15 min of treadmill gait training with Mobility Rehab. The participants were instructed to walk back and forth along a 9-m corridor at their normal, comfortable pace for 2 min. A trained research assistant placed six Opal sensors (APDM Wearable Technologies, a Clario company) on the participants’ feet, wrists, lumbar, and sternum areas. The six sensors are easy to attach, highly portable, and require no calibration or lengthy setup; enabling assessment to be performed in almost any environment. Each Opal records triaxial acceleration (±6 g), rotation (±2000 deg/s), and magnetic field (±6 gauss) at a sample rate of 128 Hz. An immediate report was generated by Mobility Lab v2 (APDM Wearable Technologies) that characterized the participants’ gait with the objective metrics, as previously published [12]. The trained research assistant determined, from the report and by their clinical judgment, which two variables (foot-strike angle, trunk coronal RoM, arm swing RoM, double-support duration, gait-cycle duration, and step asymmetry) to use for the feedback gait training on the treadmill.

#### 2.2.2. Treadmill Gait Training with Mobility Rehab

The trained research assistant monitored the participants while they trained with Mobility Rehab in the treadmill for 15 min (using the same Opals sensors), as tolerated (comfortable pace) (Figure 1A). Mobility Rehab uses the same Opal sensors of the Mobility Lab system but with new algorithms to provide real-time feedback on six objective measures (foot-strike angle, trunk coronal RoM, arm swing RoM, double-support duration, gait-cycle duration, and step asymmetry), as the user performs prescribed activities [12]. The Mobility Rehab system uses five synchronized Opal sensors, attached to the body with Velcro straps (Figure 1B). A trained research assistant placed the five Opal sensors on the participants’ feet, wrists, and sternum areas. Mobility Rehab has a tablet user interface to enable the PT to collect the data and simultaneously display movement information to their patients. The graphical user interface displays the PT-selected metrics in real time (Figure 1C), with step-by-step resolution, as shown in the screenshot of the display in Figure 2. The PT can see left and right side (top panel and bottom panel) step metrics, with each bar representing the value of one step and the grey area represents the normative values. In addition, intervals can be set as target movements (blue lines) to reach while walking on the treadmill (Figure 2). Each time a new step is added, a new bar is appended and appears on the graph. The system also records each metric used during the training session and length of the session.

The trained research assistant using the Mobility Rehab system underwent training on how to effectively use the system (e.g., sensors placement, software navigation, selection and interpretation of metrics, etc.) before data collection started. The trained research assistant designed a treatment plan for each participant to use during the 15 min of treadmill gait training with Mobility Rehab. The trained research assistant chose one or two gait metrics for the participants to focus on during the training. Each participant received visual feedback about the gait metric(s) from an electronic tablet mounted on the treadmill (Figure 1A). The display showed the participant’s gait metrics relative to age-specific normative values, that were previously collected from 120 healthy subjects between the ages of 60 and 89 years during a 2 min overground walk at a comfortable pace. The trained research assistant could also select a minimum and maximum goal for a metric, based on each participant’s abilities. In addition, the system can provide the PT and patients with online reports that convey patients’ performance during the training session and a comparison to baseline gait data. At the end of the session, the participants answered a question to rate their perception of the immediate effects of one session of treadmill gait training with Mobility Rehab on their gait, using the question: “How would you rate the effect of Mobility Rehab on your walking?” (score: Terrible (−3), Moderate (−2), Mild (−1), None (0), Mild (1), Moderate (2), Excellent (3)) [16]. We quantified this scale as negative (from −3 to −1), none (0), and positive changes (from 1 to 3).

### 2.3. Data Analysis

Normality and the presence of extreme observations were assessed using the Shapiro–Wilk test and box plots, respectively. The upper and lower body gait metrics were analyzed with a magnitude-based inference using ES (Cohen’s d). The ES and confidence interval (CI) were calculated for within-group (before and after the treadmill gait training) comparison [17,18,19]. The 95% CI of the ESs were calculated using a non-central t distribution [17,19]. Positive and negative CI (i.e., not crossing zero (0)) were considered as significant. The ES has been suggested for group comparisons as it allows the determination of the magnitude of the treatment effects and the interpretation of its practical significance [19]. Effect sizes (ES) were classified as small (ES ≤ 0.49), medium (ES, 0.50–0.79), and large (ES ≥ 0.80) [19]. We calculated two-tailed Spearman correlation coefficients to explore whether changes in upper and lower body gait metrics were associated with each other. In view of the explorative nature of this analysis, we did not correct for multiple comparisons [20]. A significance level of 5% (*p* < 0.05) was adopted. All analyses were performed using SAS 9.2 (Institute Inc., Cary, NC, USA).

## 3. Results

### 3.1. Participants

The characteristics of the 10 participants are detailed in Table 1. Importantly, 90% of the participants presented mild PD severity and they scored above 80% for Activities Specific Balance Confidence scale, which indicates that they are non-fallers [21].

### 3.2. Treadmill Gait Training with Mobility Rehab Showed Moderate to Large Effect Sizes on Overground Upper and Lower Body Gait Metrics

Regardless of the two gait metrics selected to train people with PD (1. arm swing ROM and foot-strike angle; 2. step asymmetry and arm swing ROM; or 3. step asymmetry and foot-strike angle), the foot-strike angle (ES = 0.56 and 95% CI = 0.14 to 0.97), trunk coronal RoM (ES = 1.39 and 95% CI = 0.73 to 2.06), and arm swing RoM (ES = 1.64 and 95% CI = 0.71 to 2.58) showed significant and moderate-to-large ES following treadmill gait training with Mobility Rehab; see Figure 3. Step asymmetry (ES = −0.07 and 95% CI = 0.35 to −0.21) showed no significant ES following treadmill gait training with Mobility Rehab; see Figure 3. Individual values for the two gait metrics selected to train people with PD are presented in the Appendix A.

### 3.3. Treadmill Gait Training with Mobility Rehab Showed a Safe Gait Pattern on Overground Lower Body Gait Metrics

Double-support duration (ES = −0.76 and 95% CI = −1.13 to −0.38) and gait-cycle duration (ES = −0.72 and 95% CI = −0.93 to −0.51) showed significant and moderate ES following treadmill gait training with Mobility Rehab; see Figure 3.

### 3.4. Changes in Foot-Strike Angle Are Associated with Changes in Upper and Lower Body Gait Metrics following Treadmill Gait Training with Mobility Rehab

Changes in foot-strike angle were associated with changes in gait-cycle duration (*p* = 0.025, Figure 4A) and with changes in RoM of trunk coronal (*p* = 0.039, Figure 4B).

### 3.5. Distribution of Rate Perceived

Ten percent of the participants self-reported mild effects of one session of treadmill gait training with Mobility Rehab on their gait, while 60% and 30% of the participants self-reported moderate and excellent effects, respectively (Figure 4C). No adverse events were reported.

## 4. Discussion

Our preliminary results demonstrated that one session of treadmill gait training with the use of the Mobility Rehab feedback system is feasible in people with PD. The results suggest that the system can be used to modify certain gait metrics during overground gait (foot-strike angle, trunk coronal ROM, and arm swing ROM) in people with PD.

### 4.1. Treadmill Gait Training with Mobility Rehab Is Feasible for Patients with PD

The Mobility Rehab system incorporates wearable sensors to give objective measures of gait automatically and provides visual feedback of gait via an electronic tablet during treadmill locomotion. In addition, the system can be used by PT while patients walk overground. This technology can provide real-time feedback about gait quality to PT while patients walk naturally overground, that could improve the accuracy and effectiveness of PT’s feedback to his/her patients. In fact, it is difficult to visually observe the magnitude of change in certain gait metrics, for example angle at heel strike, and give feedback on it. With Mobility Rehab, PT can design gait training for each patient using a system that accurately measures gait impairments.

In this study, one session of treadmill gait training with Mobility Rehab provided to the trained research assistant and patients immediate external feedback about their ongoing gait quality while walking on a treadmill, that improved their overground walking immediately afterwards. Visual feedback about multi-joint coordination enhances motor learning and facilitates task execution with decreased motor errors [22]. Session training with sensor-based feedback may be helpful for stimulating corrective actions and promoting self-efficacy to achieve optimal performance on gait [11,22,23]. A meta-analysis has demonstrated the effects of sensors-based feedback immediate and gait training on gait and balance measures in people with and without neurological condition [10]. However, studies have been limited to improvement in lower limb-related metrics [10,11,22,23]. For example, one study reported the feasibility and effects of sensors-based feedback on the lower-limb gait metrics (e.g., gait speed) and clinical balance of people with PD [23]. This study compared 18 sessions of gait training (30 min each session) with sensors-based feedback and a smartphone application with gait training alone. In this study, both interventions improved gait speed, but only gait training with sensors-based feedback and smartphone application improved clinical balance and was feasible and well-accepted. Feasibility is the major factor for the successful implementation of sensors-based feedback training into routine practice [24].

Our feasibility study showed that one session of treadmill gait training with this novel, PT-assisted visual feedback system (Mobility Rehab) had immediate effects on foot-strike angle, trunk coronal RoM and arm swing during overground walking. A meta-analysis has demonstrated small-to-moderate ES on gait metrics (e.g., walking speed, stride and step length) after short term (e.g., one month) gait-specific training (e.g., treadmill training) in people with PD [12]. Our previous rehabilitation study [14], showed that 6 weeks of Agility Boot Camp with Cognitive Challenges intervention resulted in small to large effects sizes (ES) on foot-strike angle (ES = 0.45), trunk coronal RoM (ES = 0.45), and arm swing RoM (ES = 0.95) in people with mild-to-moderate PD. Our feasibility study showed moderate-to-large ESs on these variables after a single session. These results are significant for people with PD, as reduced foot-strike angle [25], and decreased RoM of trunk and arm swing [26] have been shown as markers of early PD.

The RoM of trunk and arm swing is responsive to levodopa [7], but foot-strike angle may not be as responsive to levodopa [25]. Dopa insensitivity of foot-strike angle may be because reduced foot-strike angle is associated with short-latency afferent inhibition (a surrogate for cortical cholinergic activity) in people with PD but not in healthy controls [4], which suggests possible non-dopaminergic pathways linked to foot-strike angle in PD. Foot-strike angle represents the characteristics of balance and gait dysfunction in people with PD, out of 24 gait metrics (e.g., gait speed, gait-cycle duration, stride length, turn velocity, RoM of the arm and trunk, anticipatory postural adjustment, postural sway). Foot-strike angle is the second most sensitive measure highly correlated with clinical dynamic balance, perceived balance confidence, and motor function related to quality of life in people with PD [3]. Thus, Mobility Rehab could be very useful in clinical settings to improve gait metrics such as foot-strike angle, as reduced foot-strike angle is an indicator of functional mobility in people with PD [3]. Importantly, only reduced foot-strike angle discriminated daily life gait metrics in people with PD compared to healthy controls [27], which suggests that daily life monitoring of foot-strike angle is sensitive to gait impairments from PD. Thus, more extensive training with Mobility Rehab, either in the clinic or at home, may improve gait quality during daily life and it should be investigated in future studies.

The RoM of the trunk and arm swing showed the largest effect sizes (ES = 1.39 and ES = 1.64) following one session of treadmill gait training with Mobility Rehab. These results are very important for people with PD, as reduced ROM of the trunk and arm swing has been shown as a marker of early PD [26]. Trunk rotation during walking plays a critical role in successful locomotion and contributes to gait stability among older people [28,29,30] and PD [31]. Axial rigidity of the hips and trunk in people with PD is associated with motor signs (e.g., poor mobility and postural instability) [31]. Reduced lateral and rotational trunk motion during gait is a sign of impaired dynamic balance and increased fall risk [28,31]. Arm swing aids dynamic stability and efficiency of gait [28,29,30] and people with PD have decreased RoM and increased asymmetry in arm swing compared to healthy controls, even when newly diagnosed [32]. Mobility Rehab is unique in providing real-time measures of arm swing and trunk lateral stability while walking, in addition to spatial and temporal gait metrics of the lower body.

### 4.2. People with PD Showed a Safe Gait Pattern after Treadmill Gait Training with Mobility Rehab

Our exploratory correlational analysis demonstrated that changes (increases) in foot-strike angle were associated with increases in gait-cycle duration (slower gait) and with increases in RoM of trunk coronal following Mobility Rehab (Figure 4). These results indicate that walking slowly with improved trunk mobility and foot-strike angle may be a safe gait pattern for people with PD in OFF medication. Lastly, 60% of the participants rated the improvement in their gait as moderate, which demonstrated that the safe gait pattern caused by Mobility Rehab during treadmill gait training had an important impact on their gait. Although this is a feasibility study, the underlying technology developed with this proposal will eventually enable users to take mobility monitoring and rehabilitation into the home and community. This could have a large impact on patients living in rural areas, whose access to quality health care is restricted by poor infrastructure [33].

This study has limitations. First, as it is a feasibility study, the sample size was small, and we did not include a control group (gait training without Mobility Rehab). Thus, future randomized controlled trials should test the efficacy of Mobility Rehab on upper and lower body gait metrics in people with PD. Second, we assessed people with PD only in their off-medication state to reflect the true disease state and minimizes medication confounding effects. Third, Mobility Rehab provides visual feedback to PT and patients during walking on the treadmill which may be harder for people with PD who have visual problems, thus, the use of a Mobility Rehab with auditory commands may help people with visual problems, which need to be tested. Fourth, the rationale behind choosing two gait metrics to train gait is due to the short duration of the gait training in the treadmill (15 min). In fact, we believe that training more than 2 gait metrics in a short time could cause some confusion for the participant, thus decreasing the training quality due to cognitive overload to pay attention in the system while walking on the treadmill. A future, longer intervention trial will address training more gait metrics.

## 5. Conclusions

One session of treadmill gait training with Mobility Rehab is feasible and showed immediate effects on gait metrics patterns (foot-strike angle, arm ROM and lateral trunk ROM), regardless of the selected gait metric to train people with PD.

## Figures and Tables

**Figure 1 sensors-23-00128-f001:**
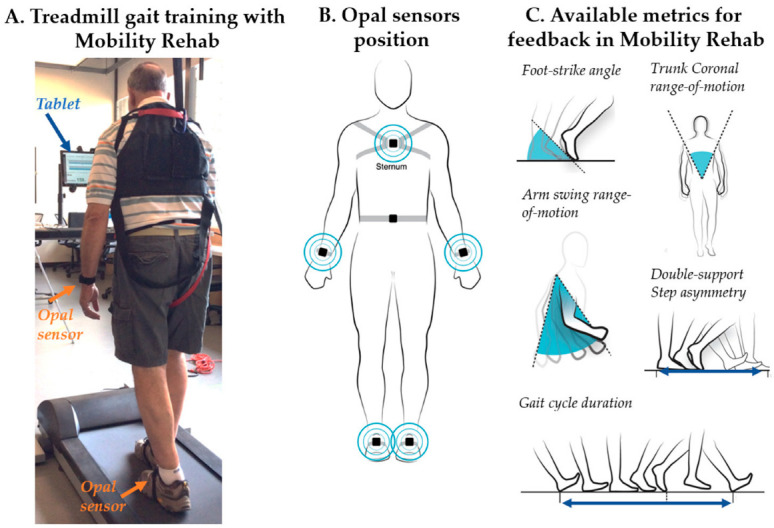
An individual with Parkinson’s disease using Mobility Rehab in the treadmill during gait training and wearing Opal sensors (inertial measurement units) (**A**). Location of the five synchronized Opal sensors attached to the body (feet, wrists, and sternum areas) with Velcro straps used during Mobility Rehab in the treadmill (**B**). Graphic of the available upper and lower body gait metrics for feedback in Mobility Rehab (**C**).

**Figure 2 sensors-23-00128-f002:**
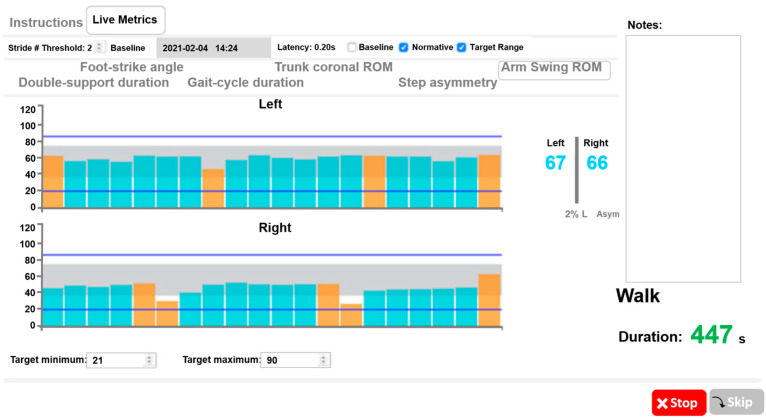
Mobility Rehab interface and visualization of gait metrics while overground walking. Example of biofeedback visualization of range-of-motion while walking. Bars indicate left and right arm range-of-motion (ROM) during straight walking (blue) and 180 degree turns (orange).

**Figure 3 sensors-23-00128-f003:**
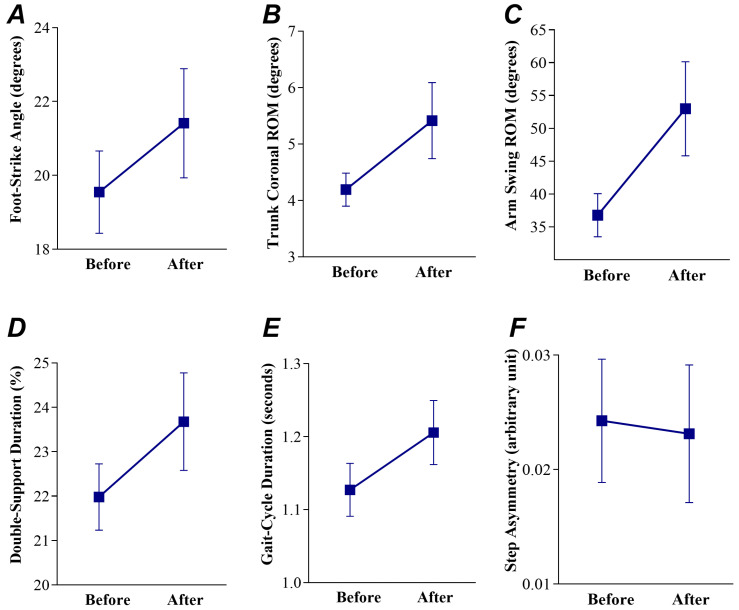
Mean and (standard error of the mean) for the foot-strike angle (**A**), trunk coronal range-of-motion (ROM) (**B**), arm swing range-of-motion (**C**), double-support duration (**D**), gait-cycle duration (**E**), and step asymmetry (**F**) before and after one session of treadmill gait training with Mobility Rehab in 10 people with Parkinson’s disease.

**Figure 4 sensors-23-00128-f004:**
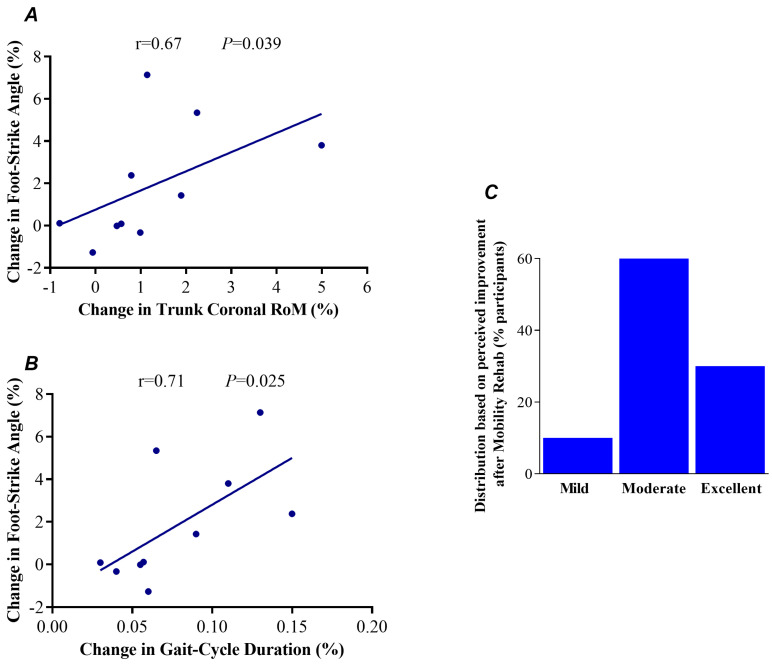
Association between the changes in foot-strike angle with the changes in trunk coronal range-of-motion (RoM) (**A**) and with the changes in gait-cycle duration (**B**) following the use of Mobility Rehab in 10 people with Parkinson’s disease. Perception rate of improvement after one session of treadmill gait training with Mobility Rehab in 10 people with Parkinson’s disease (**C**).

**Table 1 sensors-23-00128-t001:** Characteristics of the individuals with Parkinson’s disease. Means (SD) are shown.

	Participants	Range
Characteristics	(*n* = 10)	
Men/women (number)	7/3	-
Age (years)	69.3 (5.6)	60 to 75
Educational level (years)	16.0 (3.1)	12 to 20
Body mass (kg)	87.5 (2.8)	83 to 91
Height (cm)	1.7 (7.5)	1.6 to 1.8
Body mass index (kg/m^2^)	28.8 (3.0)	25 to 34
Years since diagnosis (years)	8.4 (3.7)	3 to 14
Hoehn and Yahr staging scale (a.u)		
2	9	-
3	1	-
Symptom-dominant side (R/L)	4/6	-
MDS-UPDRS-III (scores)	41.3 (6.8)	29 to 53
ABC (scores)	88.1 (5.8)	80 to 96
L-Dopa equivalent units (mg·day^−1^)	789.6 (295.3)	375 to 1200

MDS-UPDRS-III = Movement Disorders Society-Unified Parkinson’s Disease Rating Scale, motor part III; ABC = Activities-specific Balance Confidence Scale.

## Data Availability

Data available upon request.

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
