# Peer review of "Feasibility of a Novel Therapist-Assisted Feedback System for Gait Training in Parkinson’s Disease"

_sensors, 2022, doi:10.3390/s23010128_

Round 1
Reviewer 1 Report
In this paper, the authors investigate the Feasibility of a Therapist-Assisted Feedback System (Mobility Rehab) to import the walking abilities of People with Parkinson’s disease. The authors test the performance of Mobility Rehab using 2 Gait Training based on statistical methods. This topic is attractive and benefits different fields regarding motion analysis and rehabilitation. The reviewer thinks this paper can be improved according to the following comments.
Main recommendations:
1. The authors suggest their sample number is small, only 10 people with Parkinson’s disease. The reviewer knows the difficulty of finding a large number of people with a distinct disease. However, the authors can improve the sample number by dividing the experiential data into small gait motions. For example, there were 2-minute walk motions and 15 minutes of treadmill training motions. The one cycle of a gait movement averaged 2s. If the authors can divide the motion data into smaller gait motions, which can improve the sample number, achieving better-desired results (p < 0.01)
2. The therapist-Assisted Feedback System used in this research needs to be described more. Moreover, not many references were cited for the therapist-Assisted Feedback System. The authors can provide more information about this system itself and the research based on this system.
3. Not only the sample numbers can be thought of as a limitation. Were 2 Gait training motions enough to test their system for Parkinson’s? The limitations of their test method for the Feedback System can also be suggested by the authors.
Minor problems
Figure 2 needs more high resolutions because the text in the figure is hard to see. Please maintain that the words in the figure are the same size as the text in the paper.
Once again, the motion test data collected from people with Parkinson’s disease is valuable. The reviewer is like to see more significant results from these present data.
Reviewer 2 Report
for Gait Training in Parkinson's disease
1. The article contained a title and introduction according to the theme and the objectives. It is also necessary to include information about other devices the therapist uses for gait training.
2. The methodology must be detailed in the device's characteristics and the estimation of the time used in the study (2 min for walking, 15 min for therapy). Using 1 or 2 figures to describe the device and the Opal sensors will be better.
3. The results do not show in each patient the type of therapy used by the therapist. Therefore, it is necessary to specify the units for each patient because the graphics and results were not mentioned.
4. The discussion should add more references to compare the investigation.
5. The investigation is original. It was not present in other journals.
6. In the conclusions, the authors should specify the results discovered in their investigation.
7. On page 1 (Abstract) is necessary the addition of the significate of RoM and CI.
8. Page 2: in the introduction, the authors need to mention more devices used for gait metrics and generate a figure about the instrument used to understand the characteristics. In addition, in Experimental Design (Materials and Methods), the time specified for walking and therapy must be explicit.
9. In Materials and Method on page 3, in chapter f, Overground walking, the authors can show the colocation of Opal sensors by using a figure. To sum up, the paragraph: At the end of the session, people with PD completed the Perceived Usefulness and Ease of Use Questionnaire [15] and answered the following question: "How would you rate the effect of Mobility Rehab on your walking?". It is necessary to mention which variables use the questionary, the importance of the investigation, and the results obtained from the question made to the patients.
10. In the subtitle "Treadmill gait training with Mobility Rehab" (page 4), Figure 1 is unclear about the sensors' localization; also, Figure 2 could have better illumination. In addition, each patient uses a different treatment, and the therapist select 1 or 2 gait metrics there are not mentioned in the investigation.
11. In point 3.5, Perceived changes, Results page 8, the paragraph will be better in simple redaction because it is difficult to understand. The discussion must have more information about 5 to 10 new references.
Round 2
Reviewer 1 Report
The authors have revised the manuscript according to my reviews.